# Novel terpolymers based on methyl methacrylate with superior thermal stability and optical transparency for high-value applications

Shahrzad Jahangiri[ID]*, Ladan Gramifar

Department of Chemistry, College of Science, Malek-ashtar University of Technology, Shahin-shahr, Islamic Republic of Iran

* shkeykami@gmail.com

## Abstract

Methyl methacrylate (MMA) is a widely utilized polymer in high-value applications; however, it faces limitations due to its inherent thermal instability and moderate optical transparency. This study addresses these challenges by reporting the successful synthesis of novel terpolymers comprising MMA, 4-vinylbenzyl chloride (VBC), and N-2-methyl-4-nitrophenyl maleimide (MI) through radical solution polymerization. The incorporation of these co-monomers, particularly at optimized molar ratios (e.g., MMA/VBC/MI of 40:30:30), significantly enhances the performance of the polymer. The synthesized terpolymers underwent extensive characterization utilizing FTIR, NMR, CHNS, TGA, DSC, and UV-Vis spectroscopy, in addition to solubility and viscosity analyses. Our results indicate significant advancements in thermophysical and optical properties. Specifically, the terpolymers exhibited markedly improved thermal stability, with the P(MMA40-VBC30-MI30) sample demonstrating the highest decomposition temperature (T10) of 319°C, a substantial increase from PMMA's T10 of 221°C. The glass transition temperature (Tg) was also significantly enhanced, reaching 119°C across all terpolymer compositions, surpassing PMMA's Tg of 104°C. Furthermore, these innovative terpolymers achieved exceptional optical clarity. While P(MMA60-VBC10-MI30) displayed 98% light transmission, the optimized compositions, P(MMA50-VBC20-MI30) and P(MMA40-VBC30-MI30), remarkably reached 100% light transmission across the visible spectrum (400−800 nm), considerably exceeding PMMA's 90% transmission. This high transparency, even with the incorporation of the chromophoric maleimide unit, underscores a synergistic effect of VBC in reducing transparency loss. Additionally, the terpolymers demonstrated improved solubility in common solvents and reduced intrinsic viscosity (0.19–0.38 dL/g compared to PMMA's 0.86 dL/g), indicating enhanced processability. Despite the observed partial incorporation of MI (approximately 50% as determined by CHNS analysis), these methyl methacrylate-based terpolymers signify a significant advancement, providing co-optimized thermal stability, optical transparency, and enhanced processability.

**Data availability statement:** All relevant data are within the paper and its Supporting information files.

**Funding:** The author(s) received no specific funding for this work.

**Competing interests:** "The authors have declared that no competing interests exist.".

Their multifunctional properties render them highly promising candidates for demanding applications, particularly within the aerospace industry, where high performance, thermal resistance, and visual clarity are essential.

## 1. Introduction

Poly(methyl methacrylate) (PMMA), a thermoplastic polymer, is extensively employed across various industries, including optical lenses, display screens, and automotive components, due to its outstanding optical clarity, lightweight nature, and ease of processing [1,2]. The global demand for PMMA is anticipated to increase by 5.5% annually until 2028, driven by its expanding applications in the automotive, construction, and electronics sectors, highlighting its significant industrial relevance and contribution to materials science [3–5]. However, PMMA possesses inherent limitations, such as low heat resistance and high water absorption, which restrict its use in high-performance environments. These limitations can result in distortion, deformation, and loss of transparency, particularly under humid or elevated temperature conditions [6,7]. Although PMMA is optically clear under standard conditions, preserving this property in thermally or environmentally challenging settings-often characterized by high temperatures, UV exposure, or mechanical stress-poses a significant challenge due to the risk of surface degradation [8,9].

Despite extensive research into improving PMMA, a critical gap remains in simultaneously optimizing both thermal stability and optical transparency, particularly through the synergistic incorporation of specific aromatic monomers such as VBC and MI. While individual modifications have shown promise, achieving a co-optimized balance, especially in light of MI's reactivity and its potential impact on transparency, has largely gone unexplored. Additionally, research on the processability and viscosity behavior of PMMA modified with this unique combination of VBC and MI is limited. This study directly addresses these gaps by precisely tailoring the molar ratios of VBC and MI to develop novel terpolymers that overcome previous limitations and provide enhanced combined properties.

To address these challenges, various chemical modification strategies have been investigated to improve the thermal, physical, and optical properties of PMMA. These strategies include copolymerization with heat-resistant co-monomers and the incorporation of nanofillers, significantly broadening its industrial applications [10–16]. Notably, the copolymerization of functional aromatic monomers, such as 4-vinylbenzyl chloride (VBC), has demonstrated effectiveness in enhancing PMMA's thermal and mechanical properties due to VBC's rigid benzene ring and reactive vinyl group. Similarly, N-2-methyl-4-nitrophenyl maleimide (MI) has been shown to significantly increase the glass transition temperature ($T_g$), with PMMA-co-MI copolymers achieving $T_g$ values up to 40 °C higher than that of pure PMMA [15,17,18]. The enhanced aromaticity and polarity of nitro-substituted maleimide derivatives contribute to improved thermal resistance through steric hindrance and electron-withdrawing effects [19].

Previous research underscores the advantages of incorporating aromatic monomers; for instance, a PMMA-VBC copolymer demonstrated a glass transition temperature (Tg) of 105 °C [20,21]. In contrast, rigid monomers such as N-cyclohexylmaleimide exhibited a Tg of 122.54 °C and a decomposition temperature of approximately 343.40 °C [7]. Furthermore, copolymerization with phosphorus-bearing monomers, including 1,3-bisdiphenylene-2-phenylallyl (BDPA), diethyl phthalate (DEP), and methacryloyloxydecyl dihydrogen phosphate (MDP), has enhanced both Tg and flame retardancy while preserving high optical transparency [22].

Although PMMA is inherently transparent, its long-term optical stability can be affected by impurities and photodegradation [23]. Certain modifications, such as N-aryl itaconimides, maintain transparency but may contribute to increased haze [20]. Conversely, maleimide derivatives such as N-phenylmaleimide (PMI) have been found to diminish optical clarity. For instance, the incorporation of 5 mol% PMI resulted in slight turbidity and increased light scattering [7]. These findings highlight the imperative for an optimized balance between thermal stability and optical clarity in PMMA systems for multifunctional applications [24].

This research seeks to synthesize novel poly(methyl methacrylate) (PMMA)-based terpolymers utilizing methyl methacrylate (MMA), 4-vinylbenzyl chloride (VBC), and N-2-methyl-4-nitrophenyl maleimide (MI) at precisely controlled molar ratios. We hypothesize that these specific aromatic co-monomers will synergistically enhance thermal stability and significantly improve optical transparency, thereby addressing a critical gap in the existing literature. Our methodology encompasses the meticulous synthesis of homopolymers, copolymers, and terpolymers through radical solution polymerization [14–16]. The incorporation of monomers and the resulting polymer structures are confirmed using Fourier Transform Infrared (FTIR) spectroscopy, Nuclear Magnetic Resonance (NMR) spectroscopy, and Carbon, Hydrogen, Nitrogen, Sulfur (CHNS) elemental analysis. The thermal performance is rigorously analyzed through Thermogravimetric Analysis (TGA) and Differential Scanning Calorimetry (DSC), quantifying decomposition temperatures (T10, T50), char yield, and glass transition temperatures (Tg). Optical transparency is assessed using Ultraviolet-Visible (UV-Vis) spectroscopy to measure light transmission within the visible spectrum. Furthermore, we evaluate processability and material characteristics through solubility and viscosity measurements. This study offers vital insights for the optimal engineering of advanced multifunctional polymer materials with tailored optical and thermal resistance.

## 2. Materials and methods

### 2.1. Materials

Methyl methacrylate (MMA), 4-vinylbenzyl chloride (VBC), and N-2-methyl-4-nitrophenyl maleimide (MI) were acquired from Merck Chemical Co. A commercial-grade Poly(methyl methacrylate) (PMMA) utilized in aviation was sourced from CHEME Co. (Taiwan). Additional chemicals, including maleic anhydride (MAH), benzoyl peroxide (BPO), 2-methyl-4-nitroaniline, ethyl acetate (EtOAc), sodium hydroxide (NaOH), methanol, acetone, toluene, acetic anhydride, and sodium acetate, were procured from Aldrich (Milwaukee, WI), Fluka (Buchs, Switzerland), and Riedel-deHaen AG (Seelze, Germany). All reagents and solvents were of analytical purity grade and were utilized as received without further purification. MMA was purified by distillation under a nitrogen atmosphere at 100 °C, while EtOAc was distilled over potassium hydroxide under vacuum conditions.

### 2.2. Instrument and measurement

A comprehensive array of analytical techniques was employed to characterize the synthesized polymers. Fourier Transform Infrared (FTIR) spectra were obtained using a Jasco-680 spectrophotometer (Japan) at a resolution of 4 cm$^{-1}$, with specimens prepared as KBr pellets. Proton Nuclear Magnetic Resonance ($^1$H-NMR) spectra were recorded on a 500 MHz Bruker Avance 500 instrument (Germany), utilizing deuterated chloroform as the solvent. The nitrogen (N) content of the optimized terpolymer was determined through elemental analysis conducted with a Leco CHNS-932 elemental analyzer. Thermal properties were assessed via Thermogravimetric Analysis (TGA), Differential Thermogravimetric Analysis (DTG),

and Differential Scanning Calorimetry (DSC) on a PerkinElmer system (Pyris Diamond and Pyris 6 DSC), under an argon atmosphere at a heating rate of 5 °C/min, covering the temperature range from room temperature to 700 °C. Transmission spectra of the polymer solutions were measured with a Jasco V-750 UV-Vis spectrophotometer across the 200–800 nm range. Viscosity measurements were conducted using an Ostwald viscometer at 25 °C, and X-ray Diffraction (XRD) patterns were recorded utilizing a Philips X'PERT MPD diffractometer.

## 2.3. Monomer purification

The purification of MMA involved washing it 3 times with a 1/3 volume of 10% (w/w) sodium hydroxide solution, which was followed by 3 washes with deionized water to eliminate inhibitors and activate the monomer for polymerization.

## 2.4. Polymer synthesis general procedure for radical solution polymerization

All polymerizations were carried out via radical solution polymerization and conducted under reflux conditions. Full polymerizations were performed with BPO used as an initiator at 1% (w/w) relative to the total monomer mass with ethyl acetate as the main solvent. All synthesized polymers underwent precipitation in methanol followed by drying at room temperature for purification. (Fig 1) depicts the general reaction scheme for the synthesis of the MMA/VBC/MI terpolymer.

The synthesis of poly(methyl methacrylate) (PMMA) begins with dissolving 1 g (0.01 mol) of purified methyl methacrylate (MMA) in 2 mL of ethyl acetate using a 50 mL round bottom flask. Then, 0.01 g (1% w/w) of benzoyl peroxide (BPO) is added. The solution is then refluxed in a paraffin oil bath set to 80 degrees celsius for 6 hours while being stirred magnetically. The resulting polymer is cooled, dissolved in methanol, stirred for 20–30 minutes, filtered through filter paper, and dried at room temperature.

**Fig 1. Reaction scheme.** Radical polymerization reaction of the terpolymer (methyl methacrylate/ vinylbenzyl chloride/ N-2-methyl-4-nitrophenyl maleimide) Synthesis of Poly(methyl methacrylate) (PMMA).

## 2.5. Synthesis of Poly(4-vinylbenzyl chloride) (PVBC)

The synthesis of poly(4-vinylbenzyl chloride) (PVBC) starts by combining 1 mL of VBC monomer with 2 mL ethyl acetate in a 50 mL round-bottom flask. The mixture is then added with BPO in the same concentration as previously used. This mixture is also refluxed in a paraffin oil bath and stirred magnetically, but at a temperature of 80–90 degrees celsius for 12 hours. The polymer is recovered as described previously, this time stirred for 30 minutes instead of the previously stated time.

## 2.6. Synthesis of P(MMA-VBC) copolymer

A solution containing 0.73 mL of purified MMA, 0.52 g of VBC, 2 mL of ethyl acetate, and 0.01 g (1% w/w) BPO was prepared in an adequate flask. The mixture was refluxed for 12 hours at 90°C with magnetic stirring. The initial white colloidal solution that formed upon adding ethyl acetate to the monomers cleared upon heating. Copolymer precipitation required methanol as an anti-solvent, and dry air resulted in the copolymer being produced in the copolymer's room temperature. Copolymer solubility tests showed that the product was soluble in acetone while being insoluble in toluene and methanol.

## 2.7. Synthesis of N-2-methyl-4-nitrophenyl maleimide (MI) monomer

**2.7.1. Preparation of 2-methyl-4-nitrophenylmaleamic acid.** 7 g of maleic anhydride was dissolved in 15 mL of acetone. After this, the solution containing 10 g of 2-methyl-4-nitroaniline in 15 mL of acetone was added. The flask was closed and stirred for 6 hours at room temperature. After this, water was added which was accompanied with stirring at room temperature for an hour. Maleamic acid yielded a yellow precipitate which was isolated using a Buchner funnel and dried at room temperature [17].

**2.7.2. Synthesis of N-2-methyl-4-nitrophenylmaleimide.** In a round-bottomed flask, 10 g of the synthesized maleamic acid was dissolved in 15 mL of acetic anhydride. Then, 0.3 g of anhydrous sodium acetate was added to this solution. The solution was refluxed at 90 °C for 6 hours. Upon cooling, the solution was mixed with distilled water and ice in a beaker, and this mixture was stirred for 4 hours magnetically. The reaction mixture was then filtered to remove the oily phase, leaving a creamy precipitate of N-2-methyl-4-nitrophenylmaleimide. The solid was left to dry at room temperature after filtration [17].

## 2.8. Synthesis of P(MMA/VBC/MI) terpolymers

The solution free radical polymerization process for synthesizing terpolymers was conducted for 12 hours at a temperature range of 80–90 °C. The MI molar ratio was constant at 30% of total monomer feed, but three different MMA/VBC molar ratios of 60:10, 50:20, and 40:30 were tested. Radical initiator employed BPO was used in a concentration of 1% w/w, with 3–4 mL ethyl acetate serving as solvent. Ethyl acetate was used to dissolve the appropriate amount of MI monomer, which was then placed in a fitted flask. Under magnetic stirring, VBC, MMA monomers, and 1% w/w BPO were added sequentially. Magnetic stirring was applied for thorough mixing of the reaction mixture, followed by refluxing. The brown solution obtained from the reaction was allowed to cool to room temperature, then methanol was added to initiate precipitation. The stirred mixture was maintained for 20–30 minutes to ensure full polymer precipitation. Filtration yielded a creamy product that was washed with methanol and dried at room temperature.

## 3. Results and discussion

The systematic study explores the specific impact on the thermophysical and optical properties of poly(methyl methacrylate) (PMMA) through the incorporation of its monomers 4-vinylbenzyl chloride (VBC) and N-2-methyl-4-nitrophenyl maleimide (MI). Overcoming the challenges presented by pristine PMMA, especially its intrinsic limitations in high-performance PMMA applications, required synthesis and characterization of VBC and MI. Thermal characterization results showcased systematic

increases in thermal stability and optical transparency. The unique decomposition patterns, the glass transition temperatures (Tg), light transmission, solubility, and viscosity of the synthesized homopolymers, copolymers, and terpolymers underscore foundational polymer mechanisms and concepts of polymer science.

### 3.1. Structural characterization: Confirmation of monomer incorporation

Fourier Transform Infrared (FTIR) spectroscopy and Nuclear Magnetic Resonance (NMR) spectroscopy validated the successful synthesis and incorporation of monomers into the copolymers and terpolymers. Elemental analysis (CHNS) provided quantitative insights into the elemental composition.

**3.1.1. FTIR spectroscopy.** The polymers are characterized through FT-IR spectroscopy which is showcased with the FT-IR spectra in (Fig 2). For the P(MMA-VBC) 50:50 copolymer, the presence of VBC units were confirmed through C-Cl stretching and bending peaks at 695 cm$^{-1}$ and 1267 cm$^{-1}$ respectively. C=C bonds from aromatic rings were connected to peaks at 1446 cm$^{-1}$ and 1619 cm$^{-1}$. Also, a prominent peak at 1734 cm$^{-1}$, associated with carbonyl (C=O), was due to the presence of MMA. The signals arising from both monomers suggested that both of them were structured into the copolymer [25].

For the P(MMA-VBC-MI) 60:10:30 terpolymer, FT-IR peaks at 695 cm$^{-1}$ and 1265 cm$^{-1}$ yielded additional evidence for the C-Cl bond existing in the VBC group [26]. There is an intense peak at 1726 cm$^{-1}$ which contributes to the presence of C=O due to the bond fusion from PMMA and imide moiety. The bonds associated with asymmetric and symmetric stretching for -NO$_2$ also confirmed the presence of the maleimide unit as it was recorded at 1530 cm$^{-1}$ and 1348 cm$^{-1}$. Characteristic peaks of O-CH$_3$ bounded from PMMA were also noted at 1194 cm$^{-1}$ and 1148 cm$^{-1}$. The other two terpolymers, P(MMA-VBC-MI) 50:20:30 and P(MMA-VBC-MI) 40:30:30, displayed analogous peaks which are associated with the addition of all three monomers, verifying that polymerization occurred [17,27].

**3.1.2. NMR spectroscopy.** As illustrated in Fig 3, the ¹H-NMR spectrum for the terpolymer P(MMA-VBC-MI) with a 40:30:30 ratio confirms the successful synthesis of this novel polymer. The spectrum was recorded on a Bruker Avance 400 MHz NMR spectrometer using deuterated chloroform (CDCl$_3$) as the solvent. The detailed chemical shifts and assignments are as follows:

For the VBC units, distinct signals were observed in the aromatic region, with the corresponding aromatic protons 'f' and 'g' appearing as multiplet peaks around δ 7.9–8.2 ppm. In addition, the methylene protons 'h' of VBC, adjacent to the chlorine atom, were observed at δ 3.75 ppm.

For the MMA units, the methoxy protons 'c' were identified as a singlet at δ 3.62 ppm, which may overlap with the polymer backbone methylene protons 'i'. The α-methyl protons 'a' of MMA were detected as several peaks from δ 0.87–1.04 ppm, a feature characteristic of PMMA tacticity.

For the MI units, the methyl protons 'k' were seen as a singlet at δ 2.29 ppm. Other signals from the aromatic protons 'm' and 'n' of the MI unit also appeared from δ 7.9–8.2 ppm, overlapping with the aromatic protons of the VBC units.

The presence of these distinct signals, corresponding to the MMA, VBC, and MI units, confirms the successful terpolymerization. A complete, integrated spectrum for P(MMA40-VBC30-MI30) is available in the Supporting Information, providing full details of the chemical structure and proton assignments [25].

**3.1.3. CHNS elemental analysis: Evidence for partial incorporation of MI.** The CHNS elemental analysis of terpolymer P(MMA-VBC-MI) 40:30:30 was performed to confirm monomer incorporation and elemental composition. The results are given in Table 1. The data indicate that N-2-methyl-4-nitrophenyl maleimide (MI) monomer was incompletely incorporated into the terpolymer chain. For example, the experimental percentage of nitrogen (N%) was found to be 2.7%, approximately half of the theoretical value of 5.4%. Such a large deviation of 50% is a strong indication that only 50% of the MI monomer was incorporated, which was the only source of nitrogen in the terpolymer. Partial incorporation of MI might be due to its lower reactivity under radical polymerization conditions or steric hindrance and competing reactions during terpolymerization. While experimental carbon and hydrogen content values (C = 59.3% and H = 6.9%) exhibited

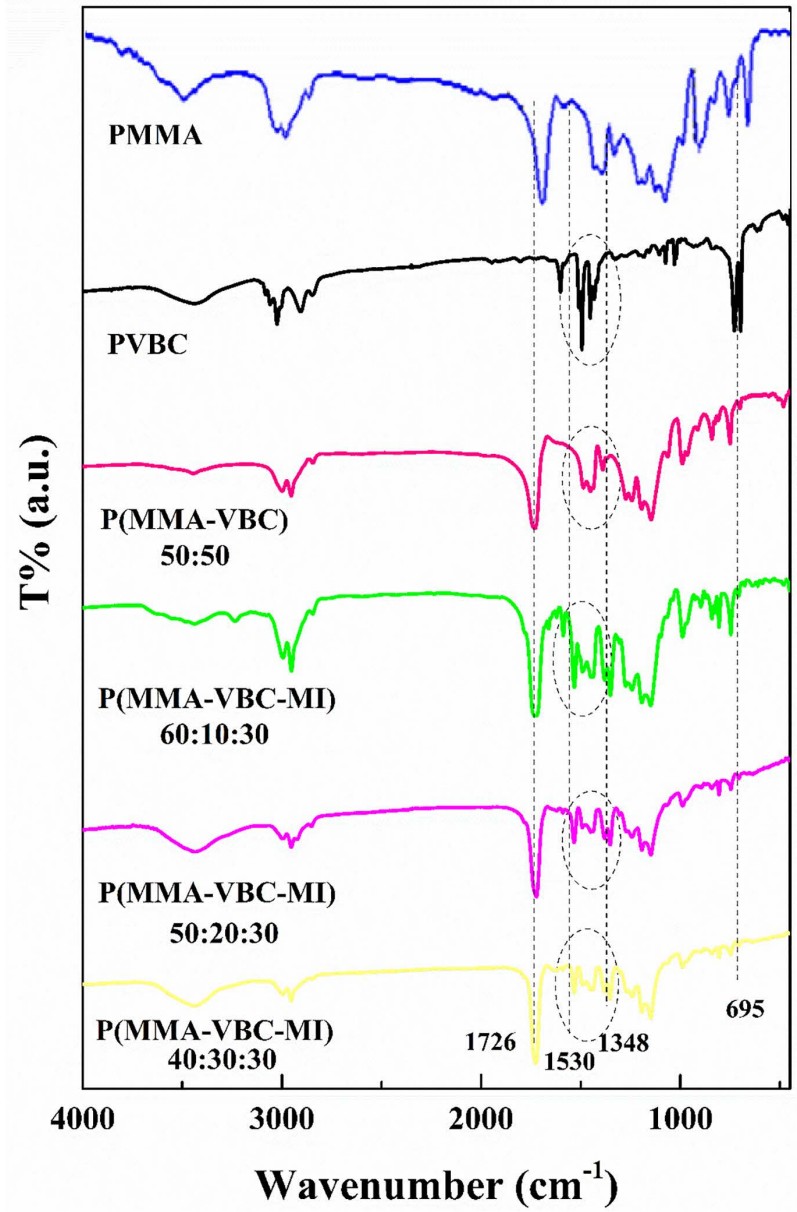

**Fig 2. FT-IR spectra of the synthesized polymers: PMMA, PVBC, P(MMA-VBC) copolymer, and P(MMA-VBC-MI) terpolymer with varying monomer ratios.**

deviation from their respective theoretical values (61.79% for C% and 5.34% for H%), the dramatic and convincing difference in nitrogen content corroborates incomplete incorporation of the MI monomer [17,27].

**3.1.4. Thermal properties: Enhanced stability from aromatic incorporation.** The materials were fully studied for thermal stability using thermogravimetric analysis (TGA/DTG) and differential scanning calorimetry (DSC) [28,29]. TGA and derivative thermogravimetric curves are shown in (Figs 4 and 5) [28]. Pure PMMA exhibited a $T_{10}$ of 221°C and a $T_{50}$ of 351°C. Thermal decomposition of PMMA happened in several steps and the peak rate of decomposition was at around 371°C, which corresponds to the DTG peak [29]. However, copolymer P(MMA-VBC) 50:50 displayed $T_{10}$ of 255°C and

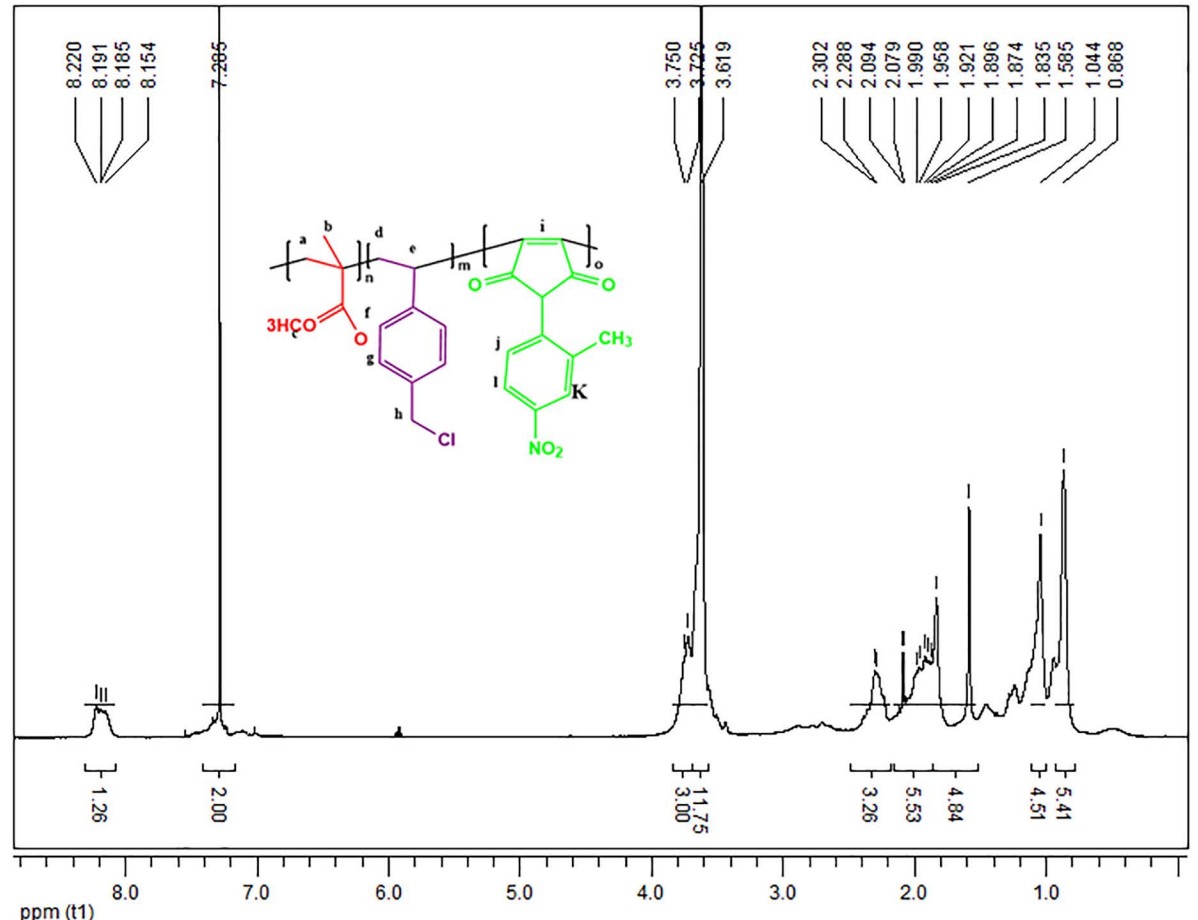

**Fig 3. H-NMR spectrum of the terpolymer P(MMA40-VBC30-MI30).**

**Table 1. Elemental Analysis (CHN) of the P(MMA40-VBC30-MI30) Terpolymer.**

| Difference (%) | | | Theoretical Value (%) | | | Experimental Value (%) | | |
|---|---|---|---|---|---|---|---|---|
| H% | C% | N% | H% | C% | N% | H% | C% | N% |
| −29.21 | 4.03 | 50 | 5.34 | 61.79 | 5.4 | 6.9 | 59.3 | 2.7 |

$T_{50}$ of 352°C. This suggests that the thermal stability of PMMA was significantly surpassed by the copolymer, indicative of improved thermal resistance due to VBC incorporation [30,31].

Crucially, the terpolymers P(MMA-VBC-MI) demonstrated steadily increasing thermal stability with rising VBC content, representing a significant advancement [30,31]. $T_{10}$ escalated from roughly 270°C for P(MMA-VBC-MI) 60:10:30–309°C for P(MMA-VBC-MI) 50:20:30 and culminating at 319°C for P(MMA-VBC-MI) 40:30:30 [31]. Similarly, $T_{50}$ values for the terpolymers varied from 338°C to 353°C [31]. The P(MMA-VBC-MI) 40:30:30 terpolymer demonstrated the highest thermal stability among all synthesized polymers, with the onset of decomposition occurring around 300°C [31]. These results not only confirm the individual contributions of VBC and MI but also highlight the synergistic potential of their combined presence in enhancing thermal performance beyond what is typically observed for PMMA or single-comonomer modifications [30,31]. A summary of the thermal properties is presented in Table 2 [31].

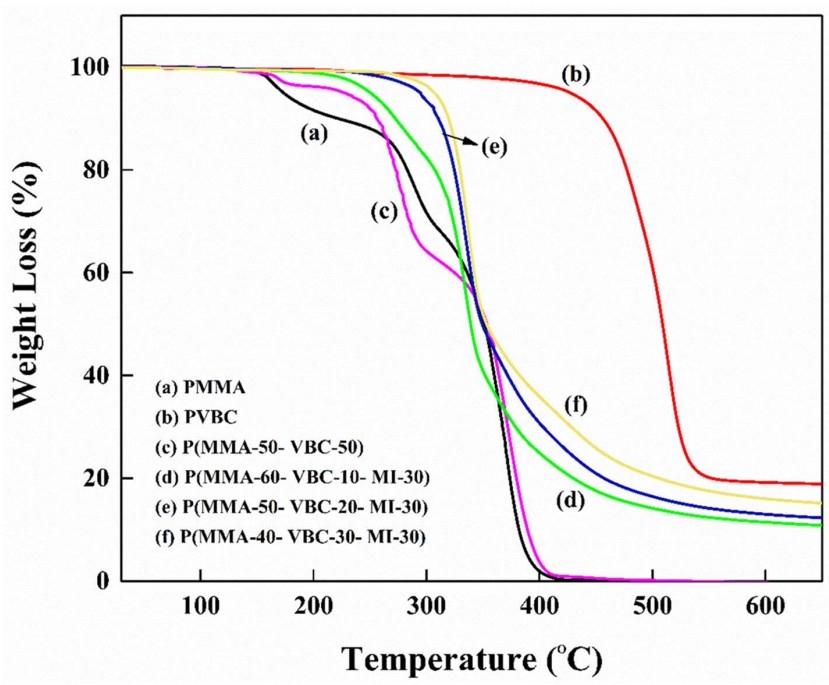

**Fig 4. Thermogravimetric analysis (TGA) curves comparing the thermal degradation behavior of the synthesized polymers: (a) PMMA, (b) PVBC, (c) P(MMA-50-VBC-50) copolymer, and P(MMA-VBC-MI) terpolymers with varying monomer ratios: (d) P(MMA-60-VBC-10-MI-30), (e) P(MMA-50-VBC-20-MI-30), and (f) P(MMA-40-VBC-30-MI-30).**

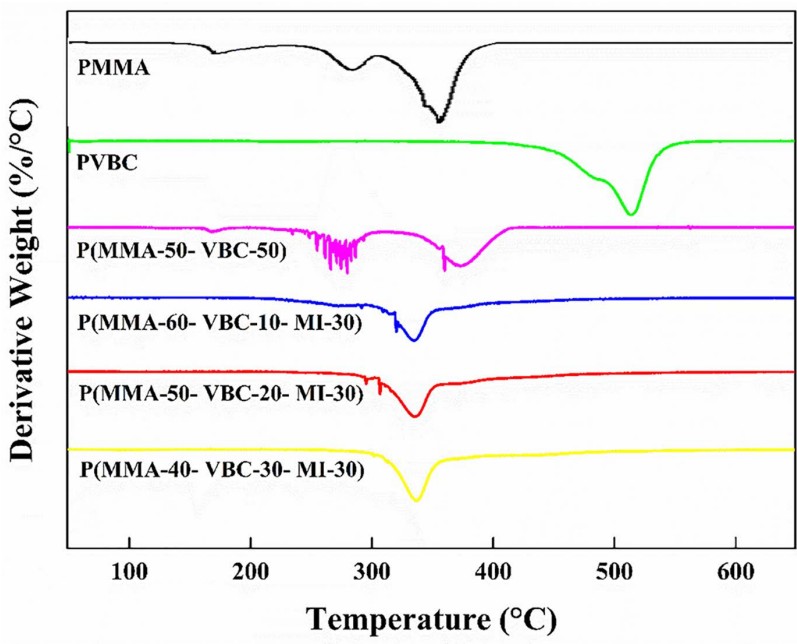

**Fig 5. Derivative thermogravimetric (DTG) curves comparing the thermal degradation rates of the synthesized polymers: PMMA, PVBC, P(MMA-50-VBC-50) copolymer, and P(MMA-VBC-MI) terpolymers with varying monomer ratios: P(MMA-60-VBC-10-MI-30), P(MMA-50-VBC-20-MI-30), and P(MMA-40-VBC-30-MI-30).**

**Table 2. Thermal Properties and Monomer Composition of Synthesized Polymers.**

| Polymer Composition | Tg (°C) | Char Yield (%) | TMax (°C) | T50 (°C) | T10 (°C) | MI% | VBC% | MMA% |
|---|---|---|---|---|---|---|---|---|
| PMMA | 104 | 8.0 | 371 | 351 | 221 | 0 | 0 | 100 |
| P(MMA) | 115 | 18.98 | 514 | 508 | 454 | 0 | 100 | 0 |
| P(MMA50-VBC50) | 119 | 0 | 372 | 352 | 255 | 0 | 50 | 50 |
| P(MMA60-VBC20-MI30) | 119 | 10 | 335 | 338 | 270 | 30 | 10 | 60 |
| P(MMA50-VBC20-MI30) | 119 | 12 | 336 | 347 | 309 | 30 | 20 | 50 |
| P(MMA40-VBC30-MI30) | 119 | 14 | 336 | 353 | 319 | 30 | 30 | 40 |

As seen in (Fig 6) and summarized in Table 2., the results from DSC explain the enhanced thermal stability [28,30]. In the DSC thermograms, an endothermic event is represented by a positive deflection (upward direction) [28]. A glass transition temperature (Tg) of 119°C was observed for the P(MMA-VBC) 50:50 copolymer and all three P(MMA-VBC-MI) interpolaters, which is in line with our findings [30]. This transition is observed as a step-like endothermic event on the curves [28]. Quite notably, this is a substantial improvement compared to the Tg of PMMA, which is 104°C, thus suggesting improved chain rigidity [29,30]. The introduction of VBC and MI monomers, with their bulky and rigid aromatic structures, greatly reduce the mobility of the polymer backbone's segments [30]. This increase in stiffness raises the amount of thermal energy required for molecular motion, thus improving thermal resistance and increasing the Tg of the material [31].

In order to improve thermal stability, the char yield of temperature rise was evaluated systematically [28,31]. Whereas PMMA showed a char yield of 8.0% at 600°C, the PVBC homopolymer offered much greater char yield close to 18.98% at the same temperature which further supports the char forming ability of the VBC unit [30]. In case of the P(MMA-VBC-MI) terpolymers, the char yields increased steadily from around 10% for P(MMA-VBC-MI) 60:10:30–12% for P(MMA-VBC-MI) 50:20:30 and further to 14% for P(MMA-VBC-MI) 40:30:30 (Table 2.) [31]. The observed increase in VBC content and the resulting residues at high temperatures demonstrate the contribution of these aromatic units to thermal stability as well as

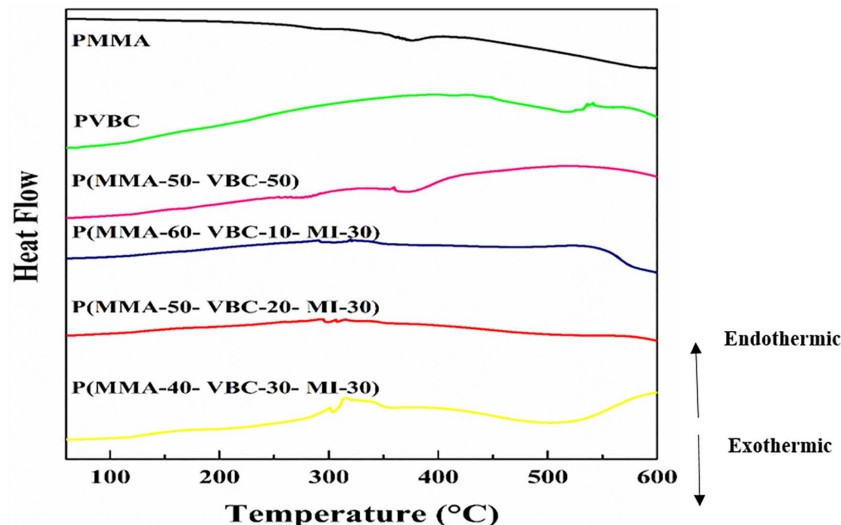

**Fig 6. Differential Scanning Calorimetry (DSC) curves comparing the thermal transitions of the synthesized polymers: PMMA, PVBC, P(MMA-50-VBC-50) copolymer, and P(MMA-VBC-MI) terpolymers with varying monomer ratios: P(MMA-60-VBC-10-MI-30), P(MMA-50-VBC-20-MI-30), and P(MMA-40-VBC-30-MI-30).** Endothermic transitions are shown as positive deflections (upward).

flame retardancy owing to charring mechanisms, which is favorable for applications needing construction materials with intrinsic fire resistance [31,32].

### 3.1.5. Optical clarity: Balancing aromaticity with transparency.

As indicated in (Fig 7), our UV-Vis spectroscopy results have shown significant enhancements in the optical transparency of the modified polymers. PMMA has been known to demonstrate 90% transmission of light in the 400nm to 800nm optical window. In addition, PVBC previously characterized as having light transmission was less than 90% by approximately 6%, indicating that PVBC, in contrast to PMMA, is less transparent. Surprisingly, the P(MMA-VBC) copolymers previously demonstrated to have 50:50 composition ratio exhibited as high as 100% transmission of light compared to PMMA, indicating its superiority. This confirms further evidence of positive optical effects with VBC incorporation in copolymer designs [33].

Although the P(MMA-VBC-MI) terpolymers incorporate maleimides, which are potential chromophores commonly linked to decreased transparency, we discovered that light transparency was surprisingly preserved, and even improved in some compositions. For example, the composition P(MMA-VBC-MI) 60:10:30 showed around 98% transmission of light. Strikingly, both P(MMA–VBC–MI) 50:20:30 and P(MMA–VBC–MI) 40:30:30 reached 100% transmission from circa 400nm and across the visible region [33].

This result is especially notable, as it defies the usual expectation that increased aromaticity through MI incorporation would necessarily result in a loss of transparency. That this was achieved with increasing VBC amounts in the terpolymers is all the more remarkable and points to a synergistic effect wherein the strategic incorporation of VBC is capable of suppressing transparency-lowering effects from the maleimide nitro group, resulting in highly transparent materials despite the inclusion of a known chromophore. The capacity to include functional aromatic units while maintaining transparency is a breakthrough for lenses, optical fibers, and displays. Such co-optimization of thermal and optical qualities is a considerable improvement over previous efforts that tended to pursue one property at the expense of the other [34].

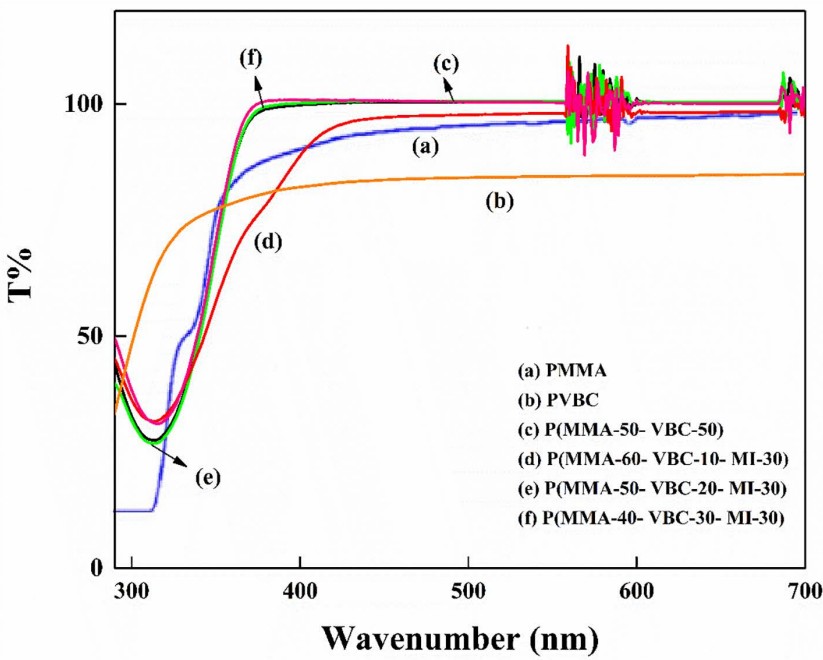

**Fig 7. UV-Vis transmittance spectra comparing the optical properties of the synthesized polymers: (a) PMMA, (b) PVBC, (c) P(MMA-50-VBC-50) copolymer, and P(MMA-VBC-MI) terpolymers with varying monomer ratios: (d) P(MMA-60-VBC-10-MI-30), (e) P(MMA-50-VBC-20-MI-30), and (f) P(MMA-40-VBC-30-MI-30).**

**3.1.6. Ancillary properties: Understanding solubility and molecular weight.** The evaluation of ancillary properties yields fundamental understanding regarding the behaviors of different materials as well as the effectiveness of their synthesis strategies.

**3.1.6.1. Solubility:** The synthesized polymers PMMA and dioxane, DMF, DMAc, EMK, and DMSO showed solubility. PMMA was insoluble in n-hexane, toluene, and distilled water, while P(MMA-VBC-MI) terpolymers P(MMA-VBC-MI showed far better solubility in chloroform, acetone, ethyl acetate, and even tetrahydrofuran. This versatility demonstrates the enhancing influence of VBC and MI units towards the solvent molecules although the oligomers must have lower critical temperature under certain conditions [27].

**3.1.6.2. Viscosity Measurements: Indicating Molecular Weight Trends:** Viscosity measurements were performed as an indirect indicator of polymer molecular weight because increased viscosity, under similar conditions, usually indicates greater molecular weight. An Ostwald viscometer was used to determine the relative viscosity ($\eta_r$) and intrinsic viscosity ($\eta_{inh}$) of the synthesized polymers over a polymer solution in ethyl acetate at 0.2 g/dL concentration at 25°C [27].

The measured viscosity values are:

- **PMMA:** $\eta_r = 1.19$, $\eta_{inh} = 0.86$ dL/g

- **P(MMA-VBC) 50:50:** $\eta_r = 1.11$, $\eta_{inh} = 0.52$ dL/g

- **Terpolymers:**

  ◦ P(MMA-VBC-MI) 60:10:30: $\eta_r = 1.04$, $\eta_{inh} = 0.19$ dL/g

  ◦ P(MMA-VBC-MI) 50:20:30: $\eta_r = 1.06$, $\eta_{inh} = 0.29$ dL/g

  ◦ P(MMA-VBC-MI) 40:30:30: $\eta_r = 1.08$, $\eta_{inh} = 0.38$ dL/g

The provided viscosity information seems to reflect a relationship with molecular weight. Concerning intrinsic viscosity, a better approximation of molecular dimensions suggests that PMMA has the greatest molecular weight of all the polymers studied. Conversely, the copolymer P(MMA-VBC) 50:50 seems to have a molecular weight that is less than that of PMMA. Among the terpolymers, P(MMA-VBC-MI) 60:10:30 appears to have the lowest molecular weight. Moreover, the increasing content of VBC in the latter terpolymers, from 10% to 30%, is associated with a comparatively greater increase in intrinsic viscosity, indicating that P(MMA-VBC-MI) 40:30:30 has much higher molecular weight than the lower VBC versions.

## 3.2. Broader implications and comparison with literature

The remarkable findings of this study include successfully optimizing the addition of VBC and MI to the PMMA matrix system, which not only increases thermal stability and optical transparency but also illustrates the complex synergistic monomer-property relationships that are essential for designing high-performance polymers. The addition of rigid aromatic components from VBC and MI into the polymers increased the glass transition temperature (Tg) to a maximum of 119°C, a clear improvement over pure PMMA's Tg of 104°C. This directly supports the established principle in polymer physics that greater molecular rigidity increases Tg. Integrating these findings with earlier works, our research corroborates and significantly builds upon the existing literature on PMMA modification [7].

As noted by Babazadeh (2006), a PMMA-VBC copolymer formed at 70 ± 1◦C for 30 hours exhibited a Tg of 105°C. While these findings were foundational, our results, which achieved a Tg of 119°C using 90◦C for 12 hours under optimized conditions, demonstrate a much greater thermal property advantage that earlier studies focused on single monomer incorporation did not fully explore. Similarly, Park et al. (2019) also synthesized PMMA copolymers with maleimide derivatives to improve heat resistance, achieving a Tg as high as 122.54°C [7,35].

Furthermore, the increase of VBC content, particularly with terpolymers achieving full light transmission, was a remarkable finding for improving optical transparency. While other researchers have also successfully improved the thermal stability of PMMA, this often comes at the cost of reduced optical clarity. A study by Atabaki et al. (2016), for instance, showed that while PMMA-co-MI copolymers had higher thermal stability, a slight decline in visible light transmission was observed with an increase in MI units. Similarly, Park et al. also observed slight turbidity in some of their copolymer films. This study's most critical and novel finding is that the strategic incorporation of higher VBC ratios (up to 30%) mitigated this negative optical effect of MI, achieving 100% transmission in terpolymers P(MMA-VBC-MI) 50:20:30 and 40:30:30 [7,17,35].

This is a significant advance because it showcases a strong synergistic effect of VBC with MI that not only increased thermal stability but also achieved exceptionally high optical transparency, thereby successfully countering the typical trade-off observed in previous work. Such integrated design strategies enable effective targeted tailoring of polymer properties, opening new avenues for PMMA in demanding applications [7].

### 3.3. Limitations and future directions

Even with the advancements obtained up to now, this study has gaps which need to be improved on while moving forward. In the conditions of free radical polymerization employed, the incorporation of MI, as evidenced by the CHNS analysis of nitrogen content, 2.7% experimental vs 5.4% theoretical, suggests MI has free radical polymerization. This incomplete incorporation process might be the reason why limited thermal enhancement was observed, assuming theoretically greater MI integration would be possible. In addition, the experimental framework was an exclusive assessment of thermophysical and optical properties with no analysis of more fundamental mechanical parameters like tensile strength, modulus, and flexibility. Most of these mechanical attributes are fundamental for practical and industrial applications. In addition to this, the controlled laboratory environment where the material was synthesized and characterized sharply contrasts with how these results would be applied industrially. In an industrial setting, parameters such as intense exposure to ultraviolet light, variation in humidity levels, and different degrees of mechanical forces would affect the functionality and material durability over an extended period.

In light of our previous studies regarding the synergistic effects of VBC and MI, we are now suggesting some new aims that can further define the field of research. Firstly, the introduction of MI into the polymers could be more precisely achieved by controlled radical polymerization (ATRP or RAFT), which could be used to further improve thermal and optical behavior by taking full advantage of MI's positive effects. Hence, other polymerization techniques need to be attempted. Secondly, the mechanical characteristics (tensile strength, impact resistance, and flexural modulus) of such novel terpolymers are also valuable not only to extend the appreciative range of their value but also due to their promising thermophysical and optical performance. Lastly, extreme testing, i.e., accelerated aging and humidity chamber testing, would further the knowledge regarding their real-world applicability and longevity, bridging gaps between laboratory findings and industry demands and pushing forward the engineering of PMMA-based materials for advanced applications.

### 4. Applications of synthesized terpolymers

Because of its outstanding optical clarity, lightweight, and ease of processing, poly(methyl methacrylate) (PMMA) is critical polymer in several sectors, particularly in aerospace. Nonetheless, its comparatively low thermal stability poses considerable difficulties in hot and demanding environments. This study addresses this critical problem by broadening the scope of PMMA in high-performance multifunctional applications. Through the controlled synthesis and characterization of modified terpolymers from methyl methacrylate (MMA), 4-vinylbenzyl chloride (VBC), and N-2-methyl-4-nitrophenyl maleimide (MI), we achieved remarkable and synergistic enhancements in the thermophysical and optical properties of PMMA. The findings indicate that these innovative terpolymers hold great promise for advanced applications, particularly in aerospace where extreme conditions require materials with exceptional multi-functional performance.

## 4.1. Enhanced thermal stability for aerospace requirements

With respect to drones, aircrafts, and spacecrafts, the aerospace industry requires polymeric materials to be a functional polymer to possess simultaneous high temperature endurance as well as mechanical strength retention throughout the polymer's service life. The synthesized terpolymers undergo extensive thermal testing, which demonstrates their purpose-fully tailored greater thermal stability compared to pure PMMA. The glass transition temperature (Tg) of PMMA is 104 °C (Table 2). The designed P(MMA-VBC-MI) terpolymers where VBC and MI are incorporated as VBC and MI structural units exhibit significantly enhanced Tg of 119 °C at all three molar ratios (MMA60:VBC10:MI30, MMA50:VBC20:MI30, and MMA40:VBC30:MI30). This Tg increase is the result of VBC and MI's rigid aromatic rings and Elevated temperature mechanical strength in aerospace applications is enhanced by the stiff structural design these polymers offer, which normally benefices these polymers.

Further to the Tg analysis, thermal decomposition kinetics ($T_{10}$) and char yield also substantiate thermal stability. For pure PMMA, $T_{10}$ is recorded at 221 °C. This is, however, significantly lower than P(MMA60-VBC10-MI30) terpolymer's 270 °C, P(MMA50-VBC20-MI30)'s 309 °C, and a peak of 319 °C at P(MMA40-VBC30-MI30). The consistent and substantial rise in $T_{10}$ with increased VBC amount strongly emphasizes the terpolymers' vast increase in thermal stability and higher operational temperature limits, directly translating to an extended lifespan and enhanced reliability of the material in critical aerospace parts. In addition, the char yield (non-volatile residue percentage at 600 °C) for the terpolymers ranged from 10% for P(MMA60-VBC10:MI30) to 14% for P(MMA40-VBC30:MI30), whereas pure PMMA was found to have 8.0% (Table 2). This increase in char yield demonstrates enhanced resistance to thermal degradation and improved carbon formation at elevated temperatures, properties that are absolutely critical for aerospace applications requiring intrinsic fire resistance and maintained structural integrity during thermal loads or fire scenarios.

## 4.2. Maintained and improved optical transparency for visual clarity

Optical transparency is a critical feature for polymers used in the aerospace industry for applications such as window cockpits, cabin displays, and sophisticated sensor housings. Poly(methyl methacrylate) or PMMA, however, is known to possess high optical transparency of approximately 90% light transmission in the visible range and 400–800 nm. A novel and remarkable feature of the synthesized terpolymers is their ability to enhance thermal stability while rigorously maintaining or improving PMMA's optic transparency. For example, the terpolymers P(MMA50-VBC20-MI30) and P(MMA40-VBC30-MI30) exhibit exceptional optical properties with 100% light transmission in the visible range. Someof the P(MMA60-VBC10-MI30) terpolymer chromophoric maleimide moiety in arms may contribute to the reduced transparency of 98%, however, the optimized ones remarkably overcome this challenge and achieve full transparency. This dual advantage within these novel polymers is important for aerospace applications which require the materials to withstand high hydrostatic pressures, extreme temperatures while maintaining exceptional visual transparency, minimal light scattering, and uncompromised thermal performance.

## 4.3. Potential as processability enhancers and functional additives

In addition to the primary benefits, the modified rheological properties of these terpolymers indicate that they could function as processability enhancers. A reduction in intrinsic viscosity (ηinh) is typically linked to improved processing, especially in intricate molding interactions. For pure PMMA, the intrinsic viscosity is 0.86 dL/g. The terpolymers, however, display much lower values: for P(MMA60-VBC10-MI30) 0.19 dL/g, for P(MMA50-VBC20-MI30) 0.29 dL/g, and for P(MMA40-VBC30-MI30) 0.38 dL/g. While viscosity is often associated with molecular weight, the lower intrinsic viscosity of these terpolymers indicates better and significantly enhanced processability in comparison to PMMA. This suggests that these terpolymers could be more easily subjected to intricate manufacturing techniques. Such enhanced processability would greatly facilitate the molding and shaping of components required in the aerospace sector and could reduce production costs while providing more design and structural complexity.

Consequently, this new type of terpolymers exhibits significant promise as self-sustaining high-performance materials, as well as additives or processing aids for commercial PMMA. These optical polymers of multifunctional nature not only offer, but simultaneously combine, improved processability, thermal stability, and optical transparency of the parts. This represents the continuing evolution of design, construction, and fabrication of polymer parts and devices for advanced aerospace systems. This is the most critical need for practical use since it guarantees high-performance materials and, at the same time, the manufacturing efficiency and economic effectiveness required by the industry.

## 5. Conclusion

This research effectively addresses significant limitations of poly(methyl methacrylate) (PMMA) by synthesizing novel copolymers and terpolymers incorporating 4-vinylbenzyl chloride (VBC) and N-2-methyl-4-nitrophenyl maleimide (MI). Our controlled synthesis has led to considerable improvements in thermal stability and optical transparency. The modified polymers demonstrate a marked increase in glass transition temperature (Tg), with all samples surpassing PMMA's Tg of 104 °C and reaching up to 119 °C. Furthermore, the terpolymers exhibit enhanced decomposition temperatures (T10) ranging from 270 °C to 319 °C. Notably, despite the incorporation of aromatic modifications, optical transparency has been significantly enhanced; terpolymers with higher VBC content achieved 100% light transmission in the visible spectrum, exceeding PMMA's 90%. This exceptional combination of properties represents a substantial advancement. We also observed a reduction in intrinsic viscosity, indicating improved processability. These findings render the new materials particularly suitable for advanced optical lenses, aerospace components, and high-performance protective coatings that demand both thermal durability and optical clarity. This work enhances the understanding of strategic monomer selection for optimizing polymer properties and paves the way for further research. Future investigations should encompass comprehensive mechanical characterization and the examination of alternative polymerization techniques to further enhance material performance.

## Supporting information

**S1 File. Supporting information.**
(ZIP)

## Author contributions

**Conceptualization:** SHAHRZAD Jahangiri.

**Data curation:** SHAHRZAD Jahangiri.

**Formal analysis:** SHAHRZAD Jahangiri.

**Writing – original draft:** SHAHRZAD Jahangiri.

**Writing – review & editing:** SHAHRZAD Jahangiri.

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
