## [Decision Letter · Decision Letter 0]

20 Aug 2025

PONE-D-25-40489Unlocking Enhanced Properties: Novel PMMA Terpolymers with Superior Thermal Stability and Optical Transparency for High-Value ApplicationsPLOS ONE

Dear Dr. Jahangiri,

Thank you for submitting your manuscript to PLOS ONE. After careful consideration, we feel that it has merit but does not fully meet PLOS ONE’s publication criteria as it currently stands. Therefore, we invite you to submit a revised version of the manuscript that addresses the points raised during the review process. Please submit your revised manuscript by Oct 04 2025 11:59PM. If you will need more time than this to complete your revisions, please reply to this message or contact the journal office at plosone@plos.org . Please include the following items when submitting your revised manuscript:

We look forward to receiving your revised manuscript.

Kind regards,

Nayan Ranjan Singha, Ph.D.

Academic Editor

PLOS ONE

2. We note that this submission includes NMR spectroscopy data. We would recommend that you include the following information in your methods section or as Supporting Information files:

1) The make/source of the NMR instrument used in your study, as well as the magnetic field strength. For each individual experiment, please also list: the nucleus being measured; the sample concentration; the solvent in which the sample is dissolved and if solvent signal suppression was used; the reference standard and the temperature.

2) A list of the chemical shifts for all compounds characterised by NMR spectroscopy, specifying, where relevant: the chemical shift (δ), the multiplicity and the coupling constants (in Hz), for the appropriate nuclei used for assignment.

3)The full integrated NMR spectrum, clearly labelled with the compound name and chemical structure.

We also strongly encourage authors to provide primary NMR data files, in particular for new compounds which have not been characterised in the existing literature. Authors should provide the acquisition data, FID files and processing parameters for each experiment, clearly labelled with the compound name and identifier, as well as a structure file for each provided dataset. See our list of recommended repositories here: https://journals.plos.org/plosone/s/recommended-repositories .

Additional Editor Comments (if provided):

Reviewers' comments:

Reviewer's Responses to Questions

**Comments to the Author**

1. Is the manuscript technically sound, and do the data support the conclusions?

Reviewer #1: Yes

Reviewer #2: Yes

2. Has the statistical analysis been performed appropriately and rigorously? 

Reviewer #1: Yes

Reviewer #2: Yes

3. Have the authors made all data underlying the findings in their manuscript fully available?

Reviewer #1: Yes

Reviewer #2: Yes

4. Is the manuscript presented in an intelligible fashion and written in standard English?

Reviewer #1: Yes

Reviewer #2: Yes

5. Review Comments to the Author

Reviewer #1: The manuscript entitled "Unlocking Enhanced Properties: Novel PMMA Terpolymers with Superior Thermal Stability and Optical Transparency for High-Value Applications" presents the synthesis, characterization and properties of PMMA-based copolymers and terpolymers via radical solution polymerization. It is an interesting study, but need corrections before publication. These are my comments

1. Modify the title, avoid abbreviation from from the title.

2. In the abstract: Include specific quantitative results to highlight key findings, simplify complex sentences, and ensure a logical flow from background to conclusion.

3. Every abbreviation should be expanded and then give its short form in the entire the manuscript

4. Revise the graphical abstract.

5. The introduction is part is too long with 1100 words. The introduction gives a thorough overview of PMMA’s industrial importance, limitations, and previous modification strategies, clearly identifying the gap in synergistically incorporating VBC and MI to enhance both thermal stability and optical clarity. However, the section could be improved by condensing repeated points on PMMA’s limitations and prior aromatic comonomer studies, shortening long sentences for clarity, highlighting the novelty earlier to engage readers before the detailed literature review, and correcting minor typographical and reference formatting issues.

6. The author should provide all chemicals used in the study, including their sources and purities. It is crucial to specify the purity of the raw materials, as the accuracy and reliability of the results depend significantly on the quality of the samples used

7. The heating rate used for the thermal properties (DSC and TGA) is missing in section 2.4

8. Avoid bulk references in the discussion section, and instead support each key finding with specific, relevant literature citations.

9. Thermal Properties: Enhanced Stability from Aromatic Incorporation. Need more reference in this section.

10. All subfigures in a given figure need to be labeled and explained in the figure title.

11. Compare the results obtained in this study can be compared with other results and hence to point out the advancement of this work.

12. The conclusion part is too long. The Conclusion should illustrate the advances and claims of innovative aspects of the research work done

13. There are few syntax errors in this manuscript

Reviewer #2: The manuscript by Shahrzad Jahangiri and Ladan Gramifar synthesized several terpolymers and copolymers of PMMA using radical solution polymerization. Characterization of the copolymers and terpolymers was performed by FTIR, NMR, CHNS, TGA, DSC, UV-Vis spectroscopy, and analyses of copolymer solution and viscosity. The terpolymers exhibit significant improvements over the PMMA in thermal stability and optical transparency. After revision, this paper can be accepted for publication in the top journal ‘PLOS One’. My comments have been listed below:

1. The molecular weights and dispersity values of the copolymers should be determined. Molecular weights of products should be obtained by using GPC.

2. In DSC termograms, which direction 'endo' or 'exo' is?

3. The Conclusions section is too long. The Conclusions section should be revised. The Conclusion section should be clear.

4. The subject is an interesting area. The manuscript probably contained novel data, but the information given is insufficient to estimate its reliability. The introduction is insufficient. It should be rewritten with clear reasoning justifying publication. The following references related to copolymerization of MMA should be added in the introduction part for a good understanding: i) Journal of Chemical Sciences 136, 68 (2024) https://doi.org/10.1007/s12039-024-02309-y ii) Chemical Papers 76, 869–878 (2022) doi: 10.1007/s11696-021-01908-0 iii) Journal of Macromolecular Science Part A 57(12), 819–825 (2020) doi: 10.1080/10601325.2020.1788393

5. The Keywords should be at most 5. Reduce the number of keywords.

6. PLOS authors have the option to publish the peer review history of their article (what does this mean? ). If published, this will include your full peer review and any attached files.

**Do you want your identity to be public for this peer review?** For information about this choice, including consent withdrawal, please see our Privacy Policy .

Reviewer #1: **Yes: ** Manammel Thankappan Ramesan

Reviewer #2: No

---

## [Author Response · Author response to Decision Letter 1]

23 Aug 2025

We have uploaded our detailed, point-by-point response to the reviewers' and editor's comments in a separate file titled "Response to Reviewers."

---

## [Decision Letter · Decision Letter 1]

29 Aug 2025

"Novel Terpolymers Based on Methyl Methacrylate with Superior Thermal Stability and Optical Transparency for High-Value Applications"

PONE-D-25-40489R1

Dear Dr. SHAHRZAD Jahangiri,

We’re pleased to inform you that your manuscript has been judged scientifically suitable for publication and will be formally accepted for publication once it meets all outstanding technical requirements.

Kind regards,

Nayan Ranjan Singha, Ph.D.

Academic Editor

PLOS ONE

Additional Editor Comments (optional):

Reviewer #1:

Reviewer #2:

Reviewers' comments:

Reviewer's Responses to Questions

**Comments to the Author**

1. If the authors have adequately addressed your comments raised in a previous round of review and you feel that this manuscript is now acceptable for publication, you may indicate that here to bypass the “Comments to the Author” section, enter your conflict of interest statement in the “Confidential to Editor” section, and submit your "Accept" recommendation.

Reviewer #1: (No Response)

Reviewer #2: All comments have been addressed

2. Is the manuscript technically sound, and do the data support the conclusions?

Reviewer #1: Yes

Reviewer #2: Yes

3. Has the statistical analysis been performed appropriately and rigorously? 

Reviewer #1: Yes

Reviewer #2: Yes

4. Have the authors made all data underlying the findings in their manuscript fully available?

Reviewer #1: Yes

Reviewer #2: Yes

5. Is the manuscript presented in an intelligible fashion and written in standard English?

Reviewer #1: Yes

Reviewer #2: Yes

6. Review Comments to the Author

Reviewer #1: The suggested comments have been appropriately addressed in the revised manuscript, making it ready for acceptance for publication

Reviewer #2: The manuscript by Shahrzad Jahangiri and Ladan Gramifar synthesized several terpolymers and copolymers of PMMA using radical solution polymerization. Characterization of the copolymers and terpolymers was performed by FTIR, NMR, CHNS, TGA, DSC, UV-Vis spectroscopy, and analyses of copolymer solution and viscosity. The terpolymers exhibit significant improvements over the PMMA in thermal stability and optical transparency. The authors fulfilled the revision of the manuscript. The manuscript was organized very well. I recommend the publication of this manuscript. It should be acceptable for publication in the top journal ‘PLOS One’.

7. PLOS authors have the option to publish the peer review history of their article (what does this mean? ). If published, this will include your full peer review and any attached files.

**Do you want your identity to be public for this peer review?** For information about this choice, including consent withdrawal, please see our Privacy Policy .

Reviewer #1: **Yes: ** M. T. Ramesan

Reviewer #2: No

---

## [Editor Report · Acceptance letter]

PONE-D-25-40489R1

PLOS ONE

Dear Dr. Jahangiri,

I'm pleased to inform you that your manuscript has been deemed suitable for publication in PLOS ONE. Congratulations! Your manuscript is now being handed over to our production team.

Kind regards,

on behalf of

Dr. Nayan Ranjan Singha

Academic Editor

PLOS ONE